

# Contribution of Cooking Emissions to the Urban Volatile Organic Compounds in Las Vegas, NV

Matthew M. Coggon.[1*], Chelsea E. Stockwell.[1,2], Lu Xu[1,2,a], Jeff Peischl[1,2], Jessica B. Gilman[1],
Aaron Lamplugh[1,2,b], Henry J. Bowman[3], Kenneth Aikin[1,2], Colin Harkins[1,2], Qindan Zhu[1,2,c],
Rebecca H. Schwantes[1], Jian He[1,2], Meng Li[1,2], Karl Seltzer[4], Brian McDonald[1], Carsten
Warneke[1]

[1] NOAA Chemical Sciences Laboratory (NOAA CSL), Boulder, CO, USA
[2] Cooperative Institute for Research in Environmental Sciences, University of Colorado Boulder,
Boulder, CO, USA
[3] Department of Physics and Astronomy, Carleton College, Northfield, MN, USA
[4] U.S. Environmental Protection Agency, Triangle Park, NC, USA
[a] now at Department of Energy, Environmental and Chemical Engineering, Washington
University in St. Louis, Missouri, USA
[b] now at Institute of Behavioral Science, University of Colorado, Boulder, CO, USA
[c] now at Department of Earth, Atmospheric and Planetary Sciences, Massachusetts Institute of
Technology, Cambridge, MA, USA

*corresponding author: matthew.m.coggon@noaa.gov

**Abstract**

Cooking is a source volatile organic compounds (VOCs) that degrades air quality. Cooking VOCs
have been investigated in laboratory and indoor studies, but the contribution of cooking to the
spatial and temporal variability of urban VOCs is uncertain. In this study, a proton-transfer-
reaction time-of-flight mass spectrometer (PTR-ToF-MS) is used to identify and quantify cooking
emission in Las Vegas, NV with supplemental data from Los Angeles, CA and Boulder, CO.
Mobile laboratory data show that long-chain aldehydes, such as octanal and nonanal, are
significantly enhanced in restaurant plumes and regionally enhanced in areas of Las Vegas with
high restaurant density. Correlation analyses show that long-chain fatty acids are also associated
with cooking emissions and the relative VOC enhancements observed in regions with dense
restaurant activity are very similar to the distribution of VOCs observed in laboratory cooking
studies. Positive matrix factorization (PMF) is used to quantify cooking emissions from ground
site measurements and compare the magnitude of cooking to other important urban sources, such
as volatile chemical products and fossil fuel emissions. PMF shows that cooking may account for
as much as 20% of the total anthropogenic VOC emissions observed by PTR-ToF-MS. In contrast,
emissions estimated from county-level inventories report that cooking accounts for less than 1%
of urban VOCs. Current emissions inventories do not fully account for the emission rates of long-



chain aldehydes reported here and further work is likely needed to improve model representations
of important aldehyde sources, such as commercial and residential cooking.
**1. Introduction**
45       Volatile organic compounds (VOCs) degrade air quality and are emitted to urban air from
many sources, including fossil fuel combustion (Warneke et al., 2012), the use of volatile chemical
products (VCPs, McDonald et al., 2018), industrial processes (Zhang et al., 2004), residential
heating and wood burning (e.g., McDonald et al., 2000;Coggon et al., 2016), cooking (Wernis et
al., 2022), and urban vegetation (e.g., Churkina et al., 2017). Each source emits a diverse set of
molecules that react alongside nitrogen oxides ($NO + NO_2 = NO_x$) to form ozone. Recent field
work in major US metropolitan areas has characterized the distribution of urban VOCs to assess
the chemical fingerprint of understudied emission sources, such as VCPs (Gkatzelis et al.,
2021b;Peng et al., 2022). These studies have shown that these sources emit oxygenated VOCs
(oVOCs) which react to form ozone and secondary organic aerosol. Models have been updated to
better describe the emissions and chemistry of select oVOCs, including alcohols, siloxanes,
glycols, and furanoids (Coggon et al., 2021;Pye et al., 2023;Qin et al., 2021).
58       Many oVOCs are emitted to urban air that have not been well-studied or incorporated into
air quality models (Karl et al., 2018). For example, McDonald et al. (2018) showed that $C > 5$
aldehydes measured in Los Angeles, CA could not be explained by emissions inventories that
contain VCPs, fossil fuels, or biogenic sources. Cooking is a source of oVOCs that is rich in
aldehydes and fatty acids (Klein et al., 2016a). Cooking VOCs have been extensively characterized
in the laboratory (Bastos and Pereira, 2010;Klein et al., 2016a;Klein et al., 2016b;Schauer et al.,
1999;Zhao and Zhao, 2018) and it has been shown that cooking is a key activity controlling the
budget of VOCs measured in indoor air (Arata et al., 2021;Klein et al., 2019;Klein et al., 2016a).
Numerous studies have shown that cooking is a ubiquitous and important component of organic
aerosol in urban areas (Hayes et al., 2013;Robinson et al., 2006;Robinson et al., 2018;Shah et al.,
2018;Slowik et al., 2010;Zhang et al., 2019) yet only a few studies have been conducted to
characterize cooking VOCs in ambient datasets (e.g., Peng et al., 2022;Wernis et al., 2022).
71       Cooking emissions result from the combustion and high temperature decomposition of
food and oils (Bastos and Pereira, 2010;Umano and Shibamoto, 1987). During heating, fatty acids
undergo thermal oxidation to produce emissions of aldehydes, ketones, alcohols, acids, and other
products of depolymerization (Bastos and Pereira, 2010;Schauer et al., 1999). The use of spices
emits monoterpenes and their derivatives (Klein et al., 2016a). Studies that have speciated VOCs
from a variety of Western cooking styles (e.g., charbroiling, grilling, frying) and ingredients (e.g.,
oils, meats, and vegetables) show that aliphatic $C_1$-$C_{11}$ aldehydes account for a large fraction of
VOCs measured by gas-chromatography and mass spectrometry (e.g., Bastos and Pereira,
2010;Klein et al., 2016b;Peng et al., 2017;Schauer et al., 1999). For example, Klein et al. (2016b)



reported that aldehydes represent > 60% of the VOC mass emitted from frying or charbroiling
meats and vegetables.
Ambient observations have shown that long-chain aldehydes are present in urban air at
significant mixing ratios (e.g., nonanal ~ 100 – 200 ppt, Bowman et al., 2003;Wernis et al., 2022).
Recent studies have used hexanal and nonanal as markers for cooking emissions in in Atlanta, GA
and Livermore, CA (Peng et al., 2022;Wernis et al., 2022). These species correlated with morning
and evening meal preparation, and it was suspected that restaurant emissions were a driving factor
for the observed temporal variability in Livermore (Wernis et al., 2022). In addition to cooking,
certain long-chain aldehydes are known to be emitted from diesel exhaust (e.g., hexanal, Gentner
et al., 2013), and some are produced from the emission and ozonolysis of oils and fatty acids
present in human skin, at the surface of ocean waters, or from other surfaces that contain
unsaturated lipids (Kruza et al., 2017;Liu et al., 2021;Wang et al., 2022;Kilgour et al., 2021).
The high abundance of aliphatic aldehydes from cooking suggests that they may be useful
markers to constrain cooking VOC emissions in urban areas. The utility of aldehydes as cooking
markers relies on the characterization of their sources in the atmosphere as well as careful
characterization of measurement techniques used to detect these species. Short-chain aldehydes
(C< 5) are unlikely to serve as useful markers because they are produced in the atmosphere from
the OH oxidation of primary organic molecules and are also directly emitted from fossil fuel
emissions and biomass burning (Gentner et al., 2013;Koss et al., 2018;de Gouw et al., 2018). Long-
chain aliphatic aldehydes (C > 6) have the potential to be useful markers for cooking in urban
areas, but their primary sources, spatial distributions, and abundances in the atmosphere remain
uncertain. Some long-chain aldehydes may be emitted from mobile sources (e.g.,Gentner et al.,
2013) while others may be emitted from surface ozone chemistry (e.g., Liu et al., 2021). No field
measurements have reported the spatial distribution of long-chain aldehydes to determine likely
sources in urban air.
This study evaluates the contribution of commercial and residential cooking emissions to
urban VOCs using mobile laboratory and ground site observations in Las Vegas, NV with
supplemental observations made Los Angeles, CA and Boulder, CO. Mobile laboratory
measurements show that long-chain aliphatic aldehydes are significantly enhanced downwind of
restaurants and exhibit spatial distributions in urban regions of Las Vegas that closely matches
restaurant density. Furthermore, these measurements show that the distribution of VOCs that
correlate with long-chain aldehydes strongly resembles the distribution observed in restaurant
plumes. We conduct a source apportionment analysis to determine the extent to which cooking
emissions impact urban VOCs relative to other important anthropogenic sources, such as motor
vehicles and VCPs, and compare these observations to commonly used emissions inventories.





**2.  Methods**

**2.1. Field Campaign Description**

Air quality measurements were performed during the 2021 Southwest Urban $NO_x$ and VOC
Experiment (SUNVEx) and Re-Evaluating the Chemistry of Air Pollutants in California (RECAP-
CA) studies. Trace gases were measured from the NOAA mobile laboratory and at ground sites
located at the Jerome Mack Air Quality Station in Las Vegas, NV and the California Institute of
Technology campus in Pasadena, CA. In Las Vegas, seven mobile laboratory drives were
conducted between 27 June and 31 July, 2021 with a focus on sampling densely populated regions.
Ground sampling was primarily conducted from 30 June– 27 July, 2021.

Figure 1 is a map of the Las Vegas valley that shows the locations of residential, commercial,
and entertainment districts. The blue dots show the locations of restaurants that are cataloged in
health inspection reports maintained by the Southern Nevada Health District (SNHD, 2022). The
wind rose is centered at the Jerome Mack Air Quality Station and shows the prevailing wind
directions and speeds observed at the ground site. The mobile laboratory sampled across the region
(e.g., Figure 3), but focused drives was conducted along the Las Vegas Strip (the purple shaded
region shown in Fig. 1). The Las Vegas Strip is an entertainment district with a high density of
casinos, hotels, bars, and restaurants that emit VCP, fossil fuel, and cooking VOCs. The Jerome
Mack ground site is located ~8km northeast of the Las Vegas Strip in a residential area with
restaurants and small commercial businesses located along major streets. The prevailing wind
patterns show that ground measurements were routinely impacted by air transported from the Las
Vegas Strip along with regions to the north with higher commercial activity.

Ground measurements during RECAP-CA were performed at the California Institute of
Technology in Pasadena, CA from 1 August – 5 September, 2021. These measurements serve as
a comparison to the observations in Las Vegas. VOCs and other trace gases were sampled from
the top of a 10 m tower. The location of the ground site was located within 0.5km from the ground
site used during the 2010 CalNex campaign (Ryerson et al., 2013). The Pasadena ground site has
been previously characterized and is a downwind receptor site for air impacted by emissions from
downtown Los Angeles, CA (e.g., de Gouw et al., 2018).

Supplemental mobile laboratory measurements were performed in Los Angeles, CA and
Boulder, CO to sample VOCs downwind of individual restaurants. These measurements serve as
comparisons against the restaurant emissions observed in the Las Vegas region.

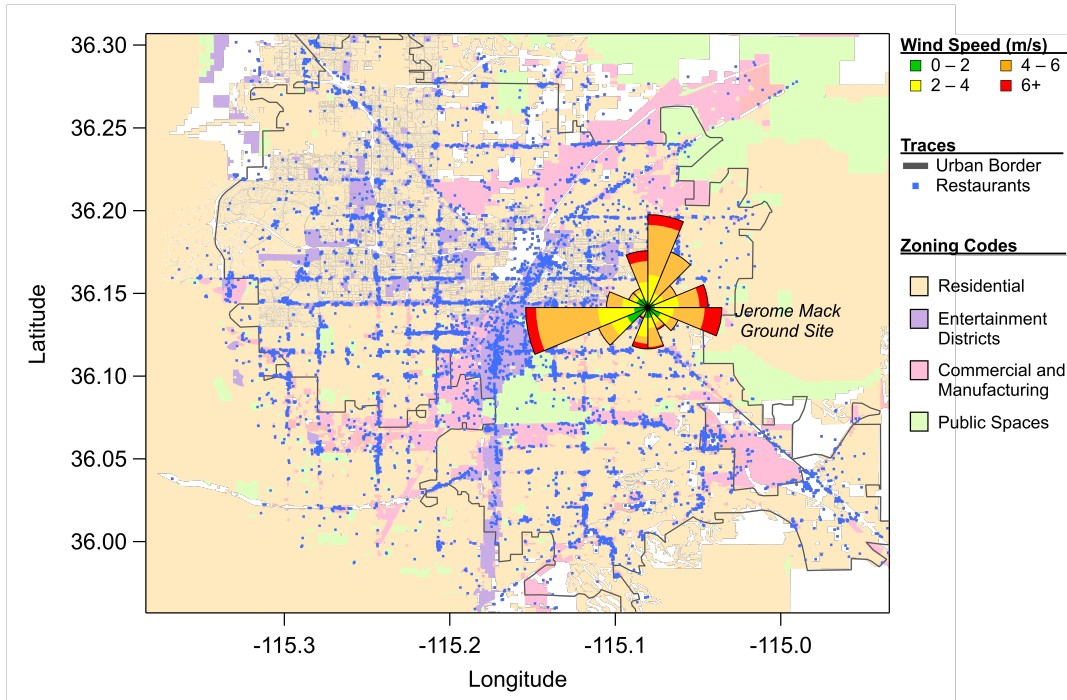

**Figure 1:** Zoning map of the Las Vegas region showing areas permitted for residential buildings, entertainment districts, commercial and manufacturing use, and public spaces. Data are available from the Clark County GIS Management Office (https://clarkcountygis-ccgismo.hub.arcgis.com, accessed September 2023) The blue dots show the locations of restaurants in the Las Vegas valley that are cataloged in health inspection reports maintained by the Southern Nevada Health District (SNHD, 2022). The wind rose shows the prevailing wind directions and speed observed at the Jerome Mack ground site. The wind rose is centered on the map at the location of Jerome Mack.

## 2.2. Instrument Description

**PTR-ToF-MS**

VOC mixing ratios were measured using a Vocus proton-transfer-reaction time-of-flight mass spectrometer (PTR-ToF-MS) (Yuan et al., 2016;Krechmer et al., 2018). The PTR-ToF-MS measures a large range of aromatics, alkenes, nitrogen-containing species, and oxygenated VOCs. A full description of the instrument during SUNVEx is provided by Coggon et al. (2023). Briefly, the PTR-ToF-MS sampled air a ~2 L min$^{-1}$ through a short ( < 1m) inlet while in the mobile laboratory. At the ground sites, the instrument sampled at ~20 L min$^{-1}$ from a 10 m tower. The Vocus drift tube was operated at 110°C with an electrical field (E) to number density (N) ratio (E/N) of 140 Td. Instrument backgrounds were determined every 2 h for ground site experiments and every ~15-30 minutes during drives by passing air through a platinum catalyst heated to 350°C.





Data were processed following the recommendations of Stark et al. (2015) using the Tofware
package in Igor Pro (WaveMetrics). The PTR-ToF-MS was calibrated using gravimetrically-
prepared gas standards for typical VOCs such as acetone, methyl ethyl ketone, toluene, and C8-
aromatics. Many compounds unavailable in gas standards were quantified by liquid calibration
methods as described by Coggon et al. (2018). This included D5-siloxane,
parachlorobenzotriflouride, octanal and nonanal. All other compounds were quantified using
estimated proton-transfer-reaction rate constants as described by Sekimoto et al. (2017). Further
corrections were applied to masses assigned to long-chain aldehydes based on observed mass-
dependent changes in fragmentation patterns described in the Supplemental Information and
shown in Fig. S2.

PTR-ToF-MS only resolves VOC molecular formula. To identify structural isomers, a
custom-built gas-chromatography (GC) instrument was used to collect and pre-separate VOCs
prior to detection by PTR-ToF-MS. A full description of the system is provided by Stockwell et
al. (2021) and its operation during SUNVEx is described by Coggon et al. (2023). Briefly, the GC
consists of a DB-624 column (Agilent Technologies, 30 m, 0.25 mm ID, 1.4 μm film thickness)
and oven identical to the system described by Lerner et al. (2017). VOCs were condensed onto a
liquid nitrogen cryotrap, flash vaporized, then passed through the column using nitrogen as a
carrier gas. The column was linearly heated during separation from 40-150°C. The effluent from
the column was directly injected in the PTR-ToF-MS inlet. At the Jerome Mack ground site, GC
samples were collected every 2 hours and immediately analyzed by PTR-ToF-MS. GC samples
were also collected during a nighttime mobile laboratory drive on July 31, 2021. These samples
were used to help interpret PTR-ToF-MS measurements along the Las Vegas Strip.

A key goal of this study is to characterize the spatial and temporal pattern of long-chain
aldehydes. Aldehydes and ketone isomers are quantified by PTR-ToF-MS using measurements of
the proton-transfer product ions (= VOC mass + $H^+$). Aliphatic aldehydes also undergo
dehydration (= VOC mass + $H^+$ – $H_2O$) and fragmentation reactions, which effectively lowers the
instrument sensitivity to the proton-transfer product. Consequently, it can be challenging to
unambiguously assign carbonyl ions to specific isomers. In the Supplemental Information, GC-
PTR-ToF-MS and mobile laboratory PTR-ToF-MS measurements show that $C_8$ and $C_9$ carbonyls
measured in urban areas are predominantly associated with octanal (detected at m/z 129,
$C_8H_{16}OH^+$) and nonanal (detected at m/z 143, $C_9H_{18}OH^+$). These ions have no detectable
interferences from ketone isomers in GC-PTR-ToF-MS spectra (Fig. S1) and the ratio of these
ions with carbonyl dehydration products most closely matches the fragmentation patterns of
aldehydes (Fig. S3). Smaller carbonyls have significant interferences from ketone isomers, which
complicates their use as markers for aldehyde emissions. Here, we focus on the spatial and
temporal trends of octanal and nonanal and use these markers to determine the fingerprint of
cooking VOC emissions.





**LGR Carbon Monoxide**

Carbon monoxide was measured at the Caltech ground site from a 10 m stainless steel tube (3.2 mm OD, 1.6 mm ID) using off-axis integrated cavity output spectroscopy (ABB Inc./Los Gatos Research model F-N$_2$O/CO-23r) (Roberts et al., 2022). Data were measured at 1-Hz and reported as 1-minute averages. Instrument precision was estimated to be ±0.2 ppb (1-σ), and the 1-σ uncertainty was estimated to be ±1% based on calibrations in the laboratory before and after the SUNVEx/RECAP-CA project.

**2.3. Positive Matrix Factorization**

Positive matrix factorization (PMF) is used to analyze the PTR-ToF-MS data and apportion VOCs to cooking and other urban sources in Las Vegas. PMF was conducted using the Source Finder (SoFi) software package in Igor Pro (Canonaco et al., 2013). Two periods are analyzed when PTR-ToF-MS measurements were available: 30 June–9 July and 19–27 July. In this analysis, we constrain PMF with a mobile source profile derived from mobile measurements following the recommendations of Gkatzelis et al. (2021b). We present a solution of factors representing mobile sources, VCPs, cooking, and regional chemical oxidation. A full description of the PMF analysis is provided in the Supplemental Information.

**3. Results**

**3.1. Long-chain aldehydes downwind of restaurants**

Previous studies have shown that cooking organic aerosols (COA) from dense restaurant clusters exhibit plume-like behavior that can impact local air quality at spatial scales of 0.5–1 km (Robinson et al., 2018;Shah et al., 2018). For example, Robinson et al. (2018) showed that organic aerosol was enhanced by as much as 100–200 µg m$^{-3}$ within 1 km of restaurants and resulted in average local organic aerosol enhancements of 3.2 µg m$^{-3}$. Consequently, it is expected that VOCs would also be significantly enhanced in close proximity of restaurants.

Figure 2 shows mobile laboratory measurements of nonanal and octanal mixing ratios downwind of two fast food restaurants in Los Angeles, CA and Boulder, CO. Mixing ratios of markers typically representative of personal care products (D5-siloxane) and motor vehicle emissions (benzene) are also shown to highlight the presence of other sources in the region. The restaurant in Los Angeles primarily serves hot dogs, while the restaurant in Boulder serves hamburgers and fried foods. The highlighted boxes show periods where the mobile laboratory was parked to sample restaurant emissions. All other data reflect sampling periods when the mobile laboratory was driven through densely populated areas of Los Angeles and Boulder. In both cases, the mobile laboratory was parked within 50m of the restaurant exhausts.


Aldehyde mixing ratios downwind of both restaurants exceeded 1 ppb. Generally, these
mixing ratios were elevated relative to the surrounding densely populated regions. Octanal and
nonanal mixing ratios were not significantly enhanced in tailpipe emissions, which is consistent
with previous studies showing that on-road emission factors of these from US vehicles are low
(Gentner et al., 2013). Octanal and nonanal were not strongly correlated with mixing ratios of D5-
siloxane, though there were periods when long-chain aldehyde and D5-siloxane enhancements
were coincident. This may result from the co-location of food and people or a possible human
emission source. Octanal and nonanal are known to be produced from skin ozonolysis (Liu et al.,
2021;Wang et al., 2022) and carbonyls are potential ingredients in fragranced consumer products,
though emissions inventories and measurements of fragrance formulations do not indicate that
octanal and nonanal are significant ingredients of VCPs (Hurley et al., 2021;McDonald et al.,
2018;Yeoman et al., 2020).

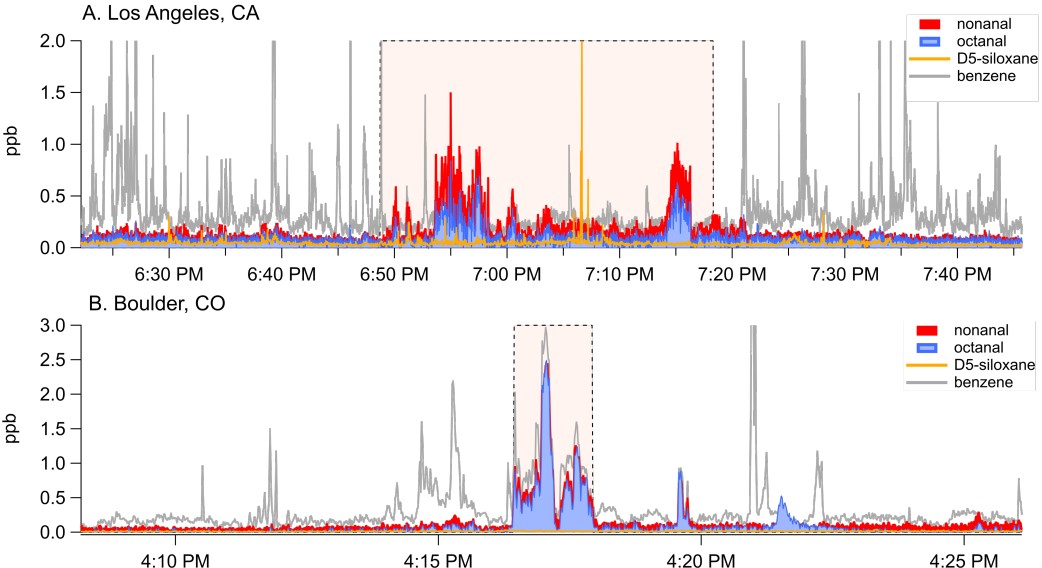

**Figure 2:** Mobile laboratory measurements of octanal, nonanal, D5-siloxane, and benzene in (A) Los
Angeles, CA and (B) Boulder, CO. The shaded regions show periods when the mobile laboratory was
parked downwind of a restaurant that primarily serves hot dogs (Los Angeles) and a restaurant that primarily
serves hamburgers (Boulder). All other data were collected while the mobile laboratory sampled air in
populated areas.

Figure 2 shows that restaurants are a strong source of aliphatic aldehydes, such as octanal and
nonanal. Based on these enhancements, it is likely that VOCs emitted from cooking are
significantly enhanced in regions with dense restaurant activity. These inferences are consistent
with previous mobile laboratory observations of primary organic aerosols. For example, Robinson





et al. (2018) found that organic aerosol in commercial districts of Pittsburgh, PA with significant
restaurant density was nearly twice as concentrated as organic aerosol in areas with highways and
significant traffic. Similar results were observed by Shah et al. (2018) in Oakland, CA where COA
constituted ~ 50% of the primary organic aerosol and was observed to be enhanced in downtown
regions where restaurant density was highest. Both studies demonstrate that air quality in urban
areas is significantly impacted by restaurant density.

Las Vegas is a sprawling city where most emission sources are concentrated along the Las
Vegas Strip (Figure 1). Figure 3 shows the spatial distribution of octanal and nonanal from all of
the mobile laboratory drives, along with a map of restaurant density calculated using the restaurant
location data shown in Figure 1. Each pixel is determined by summing the number of restaurants
over a 0.5 x 0.5 km grid. Figure 3 shows that octanal and nonanal are well-correlated ($R^2 = 0.82$)
and predominantly enhanced in the Las Vegas Strip area where anthropogenic emissions are the
highest. Figure 3A shows that this region has a high restaurant density compared to other regions
of the Las Vegas Valley. We note that brief (~1s), isolated enhancements in PTR-ToF-MS
measurements of octanal are observed outside of the Las Vegas Strip. These enhancements also
have corresponding increases in nonanal, though the ratios of these species are different than what
is observed along the Las Vegas Strip.

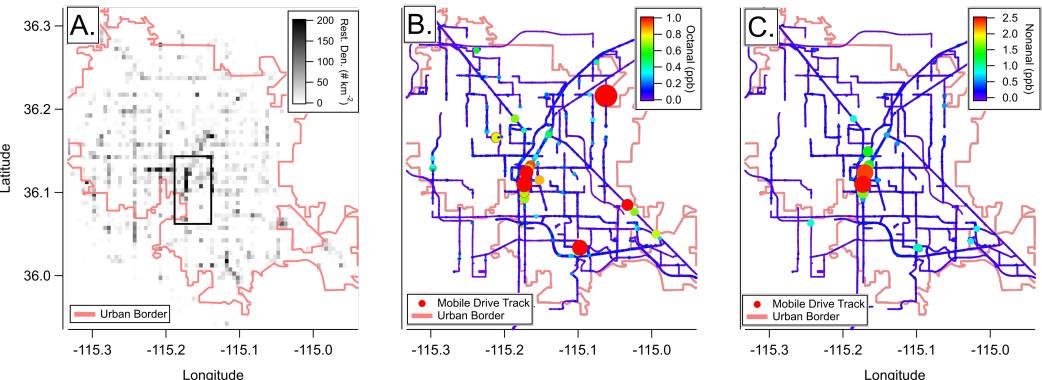


**Figure 3:** (A) Restaurant density in Clark County, NV. Restaurant density is determined using the
restaurant locations shown in Fig. 1 and is calculated by summing the number of restaurants located within
a 0.5 x 0.5 km grid. The rectangle shows the approximate region of the Las Vegas Strip. Panels (B) and (C)
show the mobile laboratory path colored by octanal and nonanal mixing ratios, respectively. Markers are
sized by the corresponding mixing ratios shown in the color scales.

Figure 4 further evaluates the spatial distribution of long-chain aldehydes by comparing
nonanal mixing ratios against the restaurant density in the proximity of the mobile laboratory.
Here, restaurant density is extracted from Fig. 3A by selecting the grid cells that are coincident
with the mobile laboratory position while sampling in the vicinity of the Las Vegas Strip (indicated



by the black box shown in Fig. 3A). The three drives shown correspond to mid-day (12:00 – 7:00 PM local time), evening (8:00 – 11:00 PM), and late-night (9:30 PM – 1:00 AM) sampling. The drives show that nonanal is generally enhanced in regions with higher restaurant density. Nonanal mixing ratios are highest during evening drives when activities from the entertainment industry in the Las Vegas Strip area, including dining, are likely most frequent (panels B and C). Enhancements in nonanal are also observed during a mid-day drive (panel A), though mixing ratios are generally lower due to a higher boundary layer and potentially lower emissions in the Las Vegas Strip area during the day. Octanal exhibits a similar relationship with restaurant density. These results are consistent with the observed enhancements in organic aerosol seen previously observed in dense restaurant regions (Robinson et al., 2018;Shah et al., 2018).

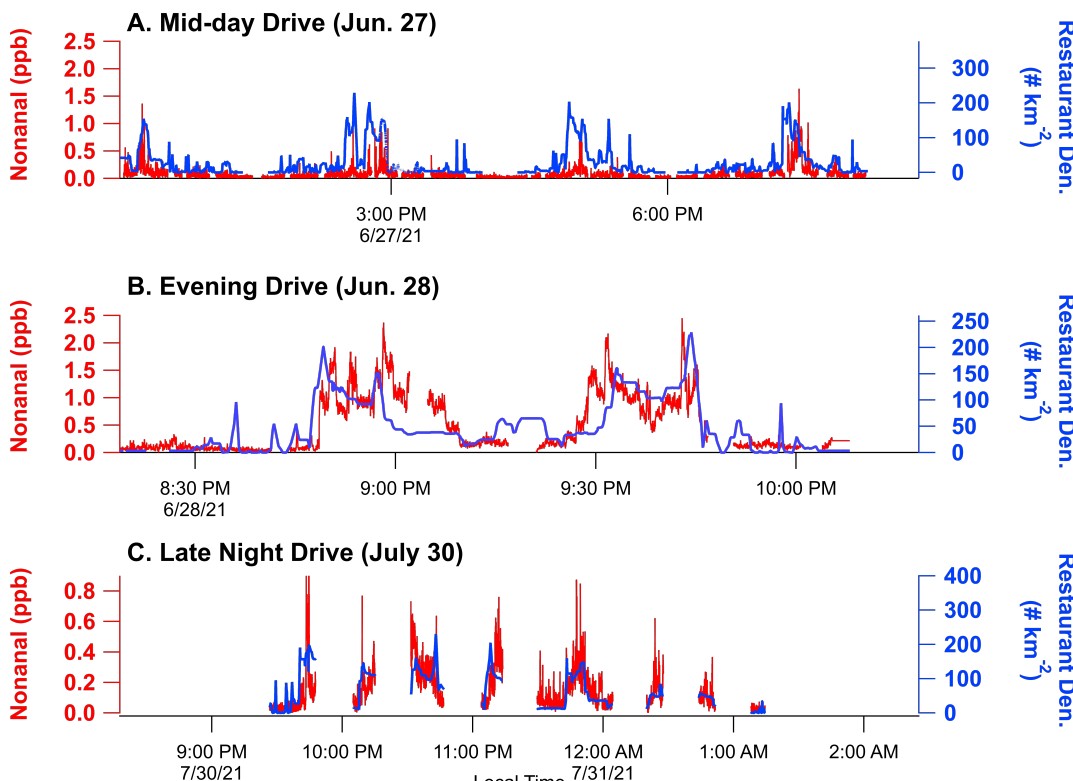

**Figure 4:** Nonanal mixing ratios and corresponding restaurant density in areas sampled by the mobile laboratory. Restaurant density is determined by averaging restaurant data in Figure 3A on a 0.5 km x 0.5 km grid, then extracting data along the mobile laboratory drive track.

Figure S12 shows that the octanal and nonanal observed along the Las Vegas Strip are also measured at the Jerome Mack ground site. The two species are well-correlated ($R^2 > 0.86$) and most abundant at night, likely due to a combination of meteorology (i.e., shallow nocturnal



boundary layer) and higher emissions in the evening (e.g., Fig. 4). Figure S12 also shows octanal
and nonanal mixing ratios observed at the Caltech ground site in Pasadena, CA. These
measurements exhibit similar temporal behavior and demonstrate that long-chain aldehyde are
ubiquitous in many urban regions. We note that nonanal ratios reported here are similar to those
observed from previous studies (~ 100 – 200 ppt, Bowman et al., 2003;Wernis et al., 2022).
**3.2. Species co-emitted with octanal and nonanal in the Las Vegas Strip**
Octanal and nonanal represent a significant fraction of cooking VOCs measured in laboratory
studies (>5%) but are just two of many VOCs emitted from cooking activities (Klein et al., 2016b).
To assess the potential fingerprint of VOCs in regions impacted by commercial cooking, we
evaluate the VOC mass spectra from the Las Vegas Strip and identify cooking VOCs by correlating
PTR-ToF-MS ions to observations of nonanal. Species are only included in this analysis if the
detected ion likely represents a proton-transfer product (i.e., fragments are excluded) and correlates
with nonanal with $R^2 > 0.8$. This high correlation coefficient is used to identify potential co-emitted
species, while excluding VOCs emitted from sources that are co-located with restaurants in the
Las Vegas Strip, such as mobile sources and VCPs. For example, D5-siloxane correlates with
nonanal with an $R^2$ of 0.78. Personal care product emissions are significant along the Las Vegas
Strip (D5-siloxane mixing ratios > 1 ppb on June 28, 2021), but octanal and nonanal have not been
reported as major components of fragranced personal care products (e.g., McDonald et al.,
2018;Steinemann, 2015;Steinemann et al., 2011;Yeoman et al., 2020;Hurley et al., 2021). A high
correlation is expected because food and people are co-located; however, in this analysis D5-
siloxane and compounds with $R^2 < 0.8$ are excluded.
Figure 5 shows the correlation spectrum of the VOCs to nonanal. The correlation coefficient
and the ratio of VOCs to nonanal are plotted versus the detected mass. The compounds that
correlate with nonanal have chemical formula of either $C_xH_yO$ or $C_xH_yO_2$ with carbon numbers
ranging between $C_2 – C_{11}$. Compounds with formula $C_xH_yO$ are likely aliphatic aldehydes with
varying degrees of saturation and some key species are highlighted for each carbon grouping
(Figure 6A). Species with the highest correlation to nonanal include octenal ($R^2 = 0.94$), decenal
($R^2 = 0.93$), butenal/crotonaldehyde ($R^2 = 0.93$), and acrolein ($R^2 = 0.92$). While individual $C_3 –$
$C_5$ species are the largest contributors to the total signal, the sum of long-chain aldehydes ($C_6 –$
$C_{11}$) are >50% of the total spectrum.
Compounds with formula $C_xH_yO_2$ likely correspond to fatty acids (Figure 5B). Gaseous acids
are released from the high temperature decomposition of long-chain acids present in meat and
previous studies have measured significant emissions of heptanoic, octanoic, nonanoic, and
decanoic acid (Schauer et al., 1999;Klein et al., 2016b). The acids in Figure 5B are some of the
most abundant acids observed along the Las Vegas Strip. The strong correlation of nonanal to fatty
acids further supports that cooking emissions are an important emitter of long-chain aldehydes in
this region.






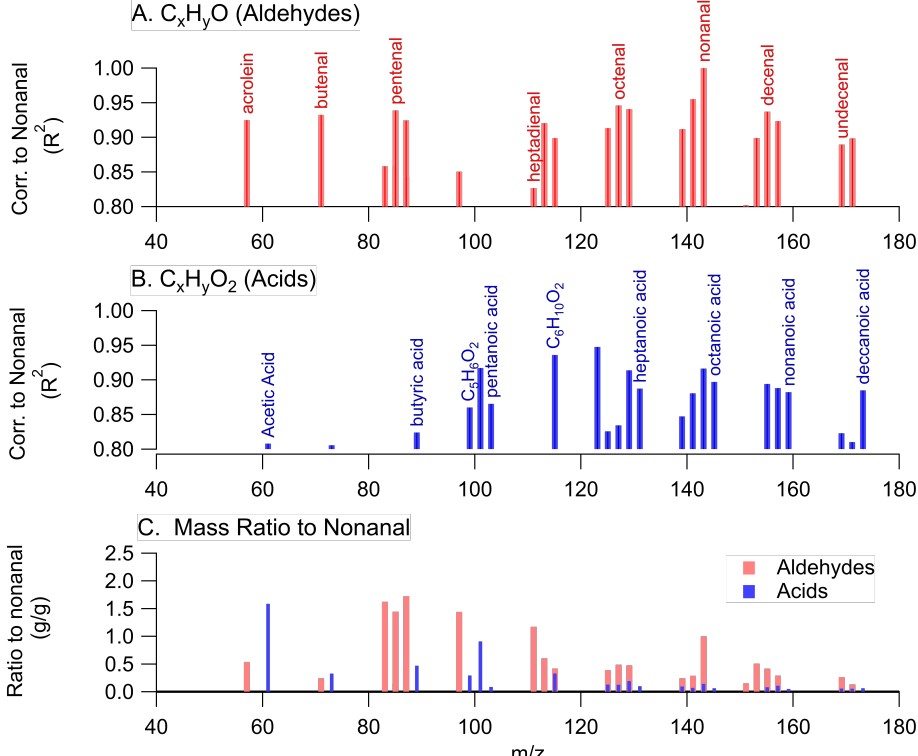

**Figure 5:** VOC correlation to nonanal in downtown Las Vegas during nighttime mobile sampling for (A)
$C_xH_yO$ species (assigned as aldehydes) and (B) $C_xH_yO_2$ species (assigned as acids). (C) VOC / nonanal
ratios for species that correlate with nonanal with $R^2 > 0.8$.


Figure 6 compares the spectra measured along the Las Vegas Strip to the spectra derived from
the individual restaurants sampled in Boulder, CO and Los Angeles, CA (Figure 2). The VOC
ratios observed in the restaurant plumes are strikingly similar to the ratios observed in downtown
Las Vegas. The Las Vegas samples most resemble those observed downwind of the hot dog
restaurant in Los Angeles. The two spectra are well correlated ($R^2 = 0.82$) which further supports
that the aldehydes and acids measured along the Las Vegas Strip were associated with restaurant
emissions. The spectra are also comparable to available PTR-ToF-MS spectra of meat cooking
emissions reported by Klein et al. (2016b). These laboratory measurements show that heptadienal,
octanal, nonanal, decadienal, and undecanal are key $C_7 - C_{11}$ aldehydes emitted when cooking
meats with vegetable oils. The same aldehydes are observed in the Las Vegas Strip area.





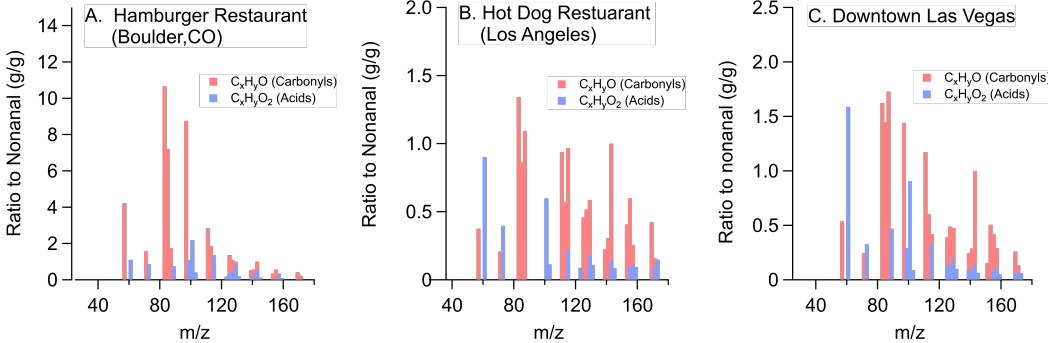

**Figure 6.** Comparison of (A) restaurant emissions from a hamburger restaurant in Boulder, CO, (B) restaurant emissions from a hot dog restaurant in Los Angeles, and (C) cooking emissions from measurements along the Las Vegas Strip.

Klein et al. (2016b) speciated laboratory cooking emissions and distinguished emissions of higher carbon aldehydes (C ≥ 7) and acids from lower carbon species. These groupings are also distinct in the laboratory VOC distributions reported by Schauer et al. (1999) and are suspected to be signatures of cooking emissions in Las Vegas (e.g., Fig. 5). Figure 7 compares the distribution of C ≥ 7 oxygenates observed in this study (top row) with the distributions reported by laboratory studies (bottom row). Schauer et al. (1999) observed that ~10% of the C ≥ 7 mass emitted from beef charbroiling was attributed to acids, while 90% was attributable to carbonyls largely composed of aldehydes. Klein et al. (2016b) observed a similar profile averaged across a range of experiments using meats and vegetable oils. In the Las Vegas Strip area, acids represented ~16% of mass associated with C ≥ 7 compounds, while the remainder is associated with carbonyls dominated by aldehydes. Carbonyls are the dominant C ≥ 7 emissions from both restaurants sampled by the mobile laboratory. The similarity in the distributions between laboratory and field observations further suggest that long-chain aldehydes are useful markers for constraining cooking emissions in urban air.





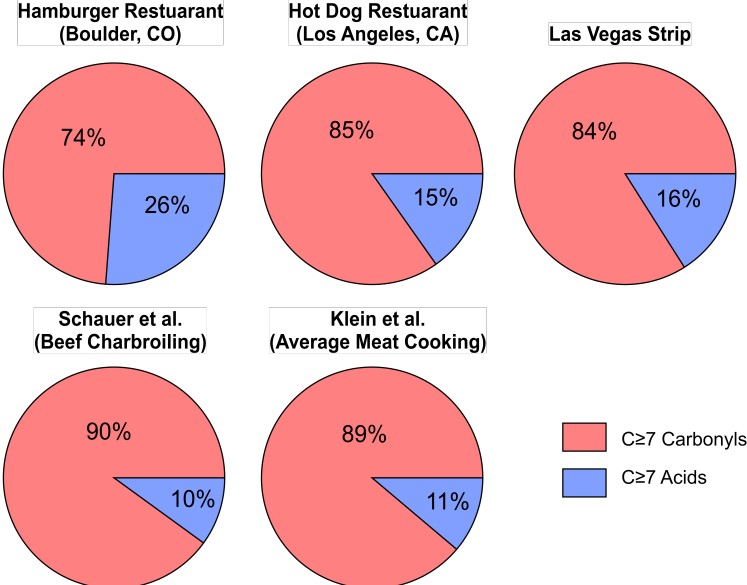

**Figure 7:** Comparison of the C ≥ 7 oxygenates measured as part of this study (top row) with those observed from laboratory experiments reported by Schauer et al. (1999) and Klein et al. (2016b) (bottom row). The distribution from Schauer et al. (1999) reflects emissions from beef charbroiling, while the distribution from Klein et al. (2016b) is derived as the average distribution from the frying of pork, chicken, beef, and fish in a range of vegetable oils.

## 4. Source apportionment from Las Vegas measurements

The analysis described in Section 3.2 provides a perspective of the key VOCs emitted from commercial cooking. Other VOCs are also likely associated with cooking but are not resolved by a simple correlation analysis due to the presence of other important sources along the Las Vegas Strip, such as ethanol from VCPs or monoterpenes from fragranced consumer products. Moreover, emissions from residential cooking may also contribute to regional VOC mixing ratios. Here, we discuss the PMF results to determine the emissions profile and the contribution of cooking to total VOC emissions observed at the Jerome Mack ground site.

Figures 8 and 9 show the PMF solution for the data collected at the Jerome Mack ground site. Figure 8 shows the time series and factor profiles, while Fig. 9 shows the average diurnal profiles. We present a 5-factor solution where VOCs are apportioned to (1) a mobile source factor, (2) a VCP-dominated factor, (3) a cooking-dominated factor, (4) a regional background plus secondary oxidation processes, and (5) a local solvent source. A full description of the PMF results is provided in the Supplemental Information.





The mobile source factor is largely composed of ethanol and $C_6$-$C_{10}$ aromatics (Fig. S4). The
VCP-dominated factor is primarily composed of ethanol (EOH), but also contains D5-siloxane,
monoterpenes, and acetone, which are common ingredients in consumer products. Both factors
resemble the solution presented by Gkatzelis et al. (2021b) for New York City. The VCP-
dominated factor derived here has two key differences from the factor derived by Gkatzelis et al.
(2021b). First, the VCP-dominated factor from New York City contained a series of other VCP
markers, including PCBTF and D4-siloxane. These molecules are largely associated with
construction activity and are expected to be emitted from the application of industrial coatings and
adhesives (Gkatzelis et al., 2021a;Stockwell et al., 2021). At the Jerome Mack ground site, PCBTF
variability was largely attributed to the local solvent factor (Fig. 8), which appeared to come from
a point source located near the ground site. Second, the VCP-dominated factor reported by
Gkatzelis et al. (2021b) also contained methyl ethyl ketone, which is a prominent solvent in
consumer and industrial VCPs. Methyl ethyl ketone was excluded from this analysis due to a water
cluster interference produced within the Vocus drift tube.
The oxidation factor is largely composed of multiple oxygenated carbon-containing masses,
which agrees with secondary factors resolved by PMF in other cities (Gkatzelis et al., 2021b;Peng
et al., 2022). At the Jerome Mack site, this factor also contained VOCs that are emitted during
daytime hours from biogenic sources (e.g., isoprene, monoterpenes) due to emissions from urban
vegetation. In general, isoprene and monoterpenes had relatively low mixing ratios over the
analyzed sampling period (< 150 ppt). For comparison, measurements in Los Angeles during
RECAP-CA show that average isoprene mixing ratios exceeded 2 ppb (Coggon et al., 2023). This
difference highlights that urban vegetation emissions in Las Vegas are significantly lower than
other cities.



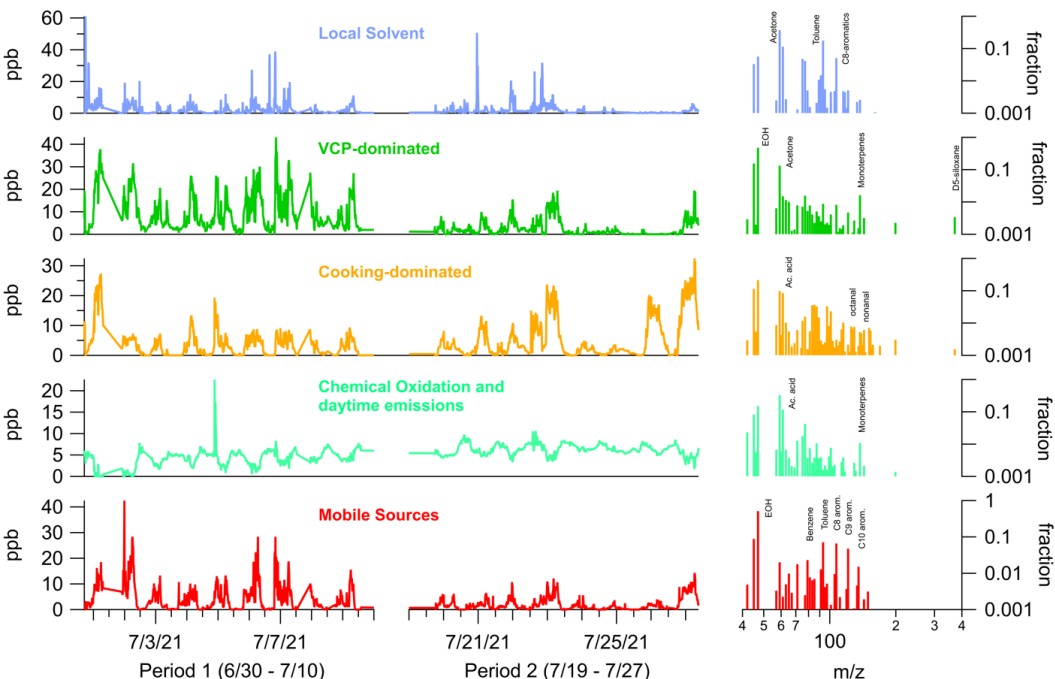

**Figure 8:** 5-factor PMF solution for the ground site data at Jerome Mack. Shown are the factor time profiles for the two time periods considered for this analysis (Period 1, 30 June–10 July and Period 2, 19–27 July) along with the resolved factor profiles.

PMF of the Jerome Mack data resolves a factor that is enriched in aldehydes, which we interpret as the cooking-dominated factor. This factor includes contributions from octanal, nonanal, acetic acid, acrolein, and higher carbon aldehydes and acids, which is consistent with the cooking emissions observed along the Las Vegas Strip and downwind of restaurants. Figure S10 compares the PMF profile to the VOC/nonanal profiles resolved by the mobile laboratory and shows that the two profiles agree for overlapping species. This agreement supports the PMF resolution of mass associated with important cooking VOCs. The PMF factor also includes ethanol, monoterpenes, and acetone/propanal, which were not resolved by the mobile laboratory analysis. Cooking emits significant amounts of ethanol and monoterpenes to indoor air (Arata et al., 2021;Klein et al., 2016a), and is a dominant source of total VOC emissions in residential indoor air (Arata et al., 2021;Klein et al., 2019). These species represent ~22% of the cooking-dominated factor.

Figure 9 shows daily mass concentrations for each factor. Gkatzelis et al. (2021b) compared the speciation profiles of the VCP-dominated and mobile source profiles to those represented in emissions inventories and found that ~53% of the mass associated with mobile source emissions and ~50% of the mass associated with VCPs results from emissions that cannot be resolved by



PTR-ToF-MS (e.g., alkanes and small alkenes). The mobile source and VCP-dominated factors
shown in Figures 9 and 10 have been adjusted to account for this unresolved mass. No adjustments
are applied to the cooking-dominated, solvent, or oxidation factor as it is assumed that PTR-ToF-
MS measures the key VOCs from these sources. This may not account for mass that has been
previously reported from cooking emissions, such as alkanes or alkenes (Schauer et al., 1999).
For the mobile source, VCP-dominated, and cooking-dominated factors, mass concentrations
are highest at night when the nocturnal boundary layer is shallow. In the daytime, the boundary
layer rapidly rises to as high as 4 km (Langford et al., 2022), and VOC concentrations decrease in
response. In contrast, the chemical oxidation + daytime emissions factor increases during daytime
hours, further supporting that this factor is largely driven by secondary processes.

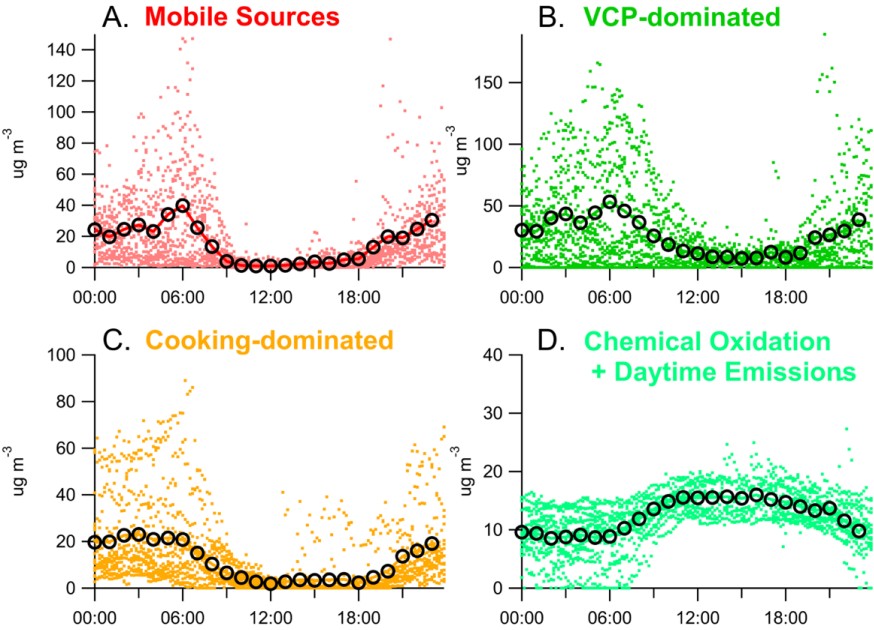

**Figure 9:** Diurnal patterns for the four factors resolved by PMF at the Jerome Mack ground site. The black
circles show the hourly mean values calculated over the full PMF solution.

To normalize for the impact of meteorology, Figure 10A shows the fraction that each primary
factor contributes total VOC mass resolved by PMF at the Jerome Mack ground site. Here, total
anthropogenic emissions are represented as the sum of the VCP-dominated, mobile source, and
cooking-dominated factors. The local solvent factor is excluded from this analysis since it is not
representative of regional VOC concentrations. Figure 10B shows the average PMF solution over
the entire analysis period.



Figure 10A shows that each factor contributes to the total anthropogenic emissions at different
times of day depending on the emission patterns. The VCP-dominated factor is the largest
contributor to total VOC emissions in Las Vegas and constitutes 40 – 80% of the primary VOCs
resolved by PMF. These concentrations are largely driven by the high emissions of solvents, such
as ethanol and acetone, which is consistent with observations from NYC (Gkatzelis et al., 2021b).
VCP emissions exhibit the highest relative abundances early in the day (~11:00 AM), then
decrease in relative abundance throughout the day. This behavior is similar to the diurnal pattern
of personal care product emissions observed in cities such as Boulder, CO where the mixing ratios
of D5-siloxane from deodorants and hair products peak during morning hours and decayed as
personal care products evaporate (Coggon et al., 2018). During evening and rush hour periods,
mobile sources constitute ∼ 30-40% of the total primary VOC mixing ratios, but then decrease
during midday due to both a large enhancement of VCPs, but also lower emissions from mobile
sources. Over the entire dataset, VCPs and mobile sources are estimated to represent 54% and 25%
of the total anthropogenic VOCs, respectively (Fig. 10B).
The cooking-dominated factor represents 10-30% of the total primary VOC mass resolved by
PMF, depending on the time of day. The relative fraction of the cooking-dominated factor peaks
in the mid-afternoon, as well as in the evening and night when activity along the Las Vegas Strip
is highest. Similar behavior has been observed in the relative abundance of primary cooking
organic aerosol in cities such as Los Angeles (Hayes et al., 2013). The cooking-dominated factor
is estimated to represent as much as 21% of the total anthropogenic VOCs over the entire dataset.
The fraction of cooking VOCs estimated here (21%) is specific to a site downwind of the Las
Vegas Strip where restaurants are abundant. These impacts are likely to vary across urban areas
based on ground site locations, restaurant density, and differences in the proportions of fossil fuel
and VCP emissions. Cooking emissions are commonly resolved from the source apportionment of
organic aerosol measurements in US cities (e.g., Hayes et al., 2013;Lyu et al., 2019;Zhang et al.,
2019;Xu et al., 2015). Far fewer studies have estimated the impact of cooking on the outdoor VOC
burden in US urban areas. Recently, cooking emissions were identified from source apportionment
of thermal desorption aerosol gas chromatograms and shown to be present at significant mixing
ratios in Livermore, CA (Wernis et al., 2022). Similarly, source apportionment of PTR-ToF-MS
data from Atlanta, GA shows that cooking emissions mixed with biomass burning were
responsible for 6–15% of the reported VOC carbon, which included contributions from fossil fuel,
VCPs, and biogenic sources (Peng et al., 2022). These proportions are similar to those reported
here, suggesting that cooking VOCs represent a significant fraction of total anthropogenic VOCs
in other US cities. Table S1 summarizes the cooking profile resolved by the PMF analysis. This
profile could be used to compare against measurements of cooking VOCs in other urban areas.



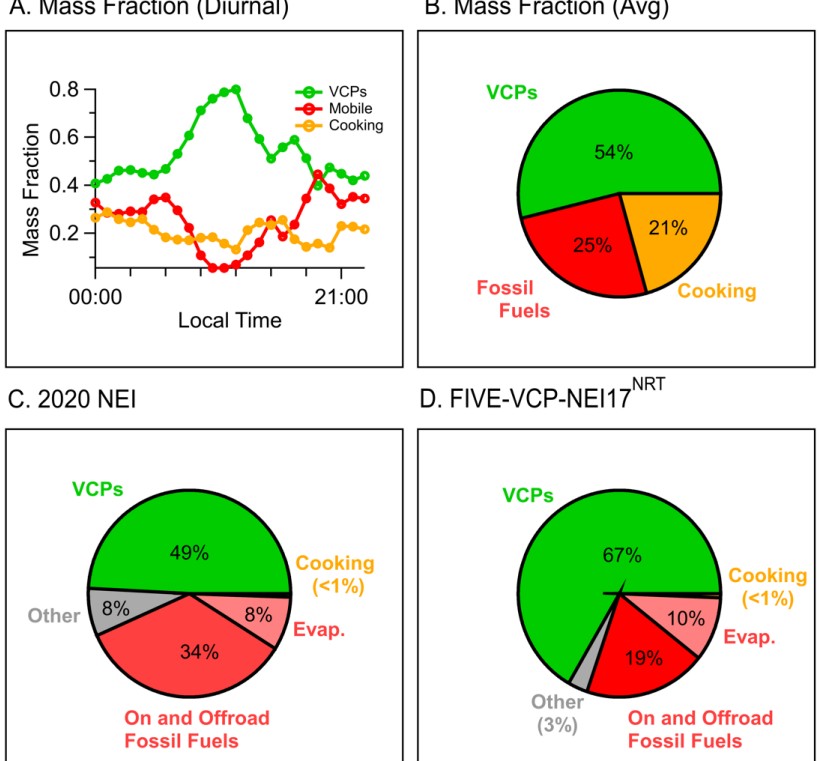

**Figure 10:** (A) Diurnal contribution of VCP, mobile sources, and cooking factors to the sum of primary emissions apportioned by PMF (= VCP + mobile source + cooking). (B) Average contribution of the VCP, mobile source, and mobile source factors to the PMF solution. (C) Distribution of anthropogenic VOC emissions from the 2020 National Emissions Inventory for Clark County, NV. (D) Distribution of anthropogenic VOCs from FIVE-VCP-NEI17[NRT] in Clark County, NV. The "other" category in both inventories reflect emissions from industry, farming, and electric power generation.

## 5. Comparison to Inventory Emissions

The PMF results shown in Figure 10B are compared against the distribution of anthropogenic VOCs reported in emissions inventories used to model air quality (Figure 10, panels C and D). Panel C shows emissions reported in the 2020 National Emissions Inventory (NEI) for Clark County, NV. The NEI is a benchmark for determining US emissions standards and its methodology is fully described by the US Environmental Protection Agency (EPA, 2023). The NEI for Clark County includes emissions for mobile sources (e.g., on- and offroad vehicles), fossil fuel evaporative sources (e.g., gasoline stations), solvent evaporative sources (e.g., VCPs), and miscellaneous point and area sources. Cooking emissions in the NEI predominantly result from commercial sources with minor contributions from residential backyard barbequing.



566

567 Panel D shows the distribution of VOCs represented by the FIVE-VCP-NEI17[NRT] inventory
568 described by He et al. (2023). The FIVE-VCP inventory was developed following the methods
569 prescribed by McDonald et al. (2018) and was recently used to determine VCP impacts on air
570 quality in US cities  (e.g., Coggon et al., 2021;Qin et al., 2021). He et al. (2023) updated FIVE-
571 VCP to include 2017 NEI emissions (NEI17) and revised VOC emissions with near real-time
572 (NRT) adjustment factors to account for COVID-19 impacts on various emission sectors. Mobile
573 source emissions are determined from fuel sales and on-road and off-road emission factors. VCP
574 emissions are estimated based on the mass balance of the chemical product industry for 2010 then
575 adjusted to 2021 emissions based on the long-term declining trends in VCP emissions reported by
576 Kim et al. (2022) and economic scaling factors reported by He et al. (2023). Other sectors are from
577 the NEI17 and are similarly updated with near real-time adjustment factors. Cooking emissions in
578 FIVE-VCP-NEI17[NRT] are the same as those used in the 2017 NEI.

579

580 The 2020 NEI and FIVE-VCP-NEI17[NRT] inventories both indicate that VCPs are the dominant
581 VOC emission sources in Clark County. Fossil fuel emissions are the next largest source, though
582 differences between the two inventories are evident. In the NEI, total fossil fuel emissions ( = on-
583 and offroad emissions + evaporative emissions) are 15% lower than VCP emissions. In FIVE-
584 VCP-NEI17[NRT], fossil fuel emissions are 57% lower than VCPs. These differences are reflected
585 in previous comparisons between the NEI and FIVE-VCP (Coggon et al., 2021;McDonald et al.,
586 2018). The PMF solution shows that fossil fuels are ~54% lower than VCPs, which is most
587 consistent with FIVE-VCP-NEI17[NRT]. Zhu et al. (2023) show that FIVE-VCP speciation agrees
588 well with VOCs primarily emitted from fossil fuel and VCPs reported during SUNVEx and
589 RECAP-CA. Aldehydes, ethanol, and monoterpenes were underestimated which may point to the
590 importance of missing emission sources, such as cooking.

591

592 The fraction of total cooking VOC emissions represented in both inventories is significantly
593 lower than the fraction resolved by PMF. Commercial sources dominate the cooking emissions in
594 the 2017 and 2020 NEI and are estimated based on food consumption estimates and emission
595 factors derived from laboratory studies (e.g., Schauer et al., 1999). The differences between the
596 PMF results and what is reported in the inventories may be partially explained by the spatial scale
597 of the datasets – the inventories represent county-level emissions estimates, while the observations
598 are specific to a site strongly influenced by the Las Vegas Strip. Nevertheless, the magnitude of
599 the disconnect highlights the need for further analysis. It is possible that the emission factors and/or
600 consumption of oils, meats, and other foods are different from what is reflected in laboratory
601 studies.


603 **6. Conclusions**




Mobile laboratory and ground site measurements were analyzed to determine the importance
of cooking emissions on urban VOC composition in Las Vegas, NV. PTR-ToF-MS data show that
cooking is a significant source of long-chain aldehydes to urban air. Measurements of octanal and
nonanal are found to be useful markers to evaluate cooking emissions due to their abundance in
restaurant plumes and local enhancements in areas with high restaurant density. A comparison of
the mass spectra downwind of restaurants to those obtained in regions with significant commercial
cooking show similar distributions in aldehydes and fatty acids known to be emitted from
laboratory cooking experiments.

Based on a PMF analysis, it is estimated that cooking emissions represent as much as 20% of
the anthropogenic VOCs emitted to the atmosphere in Las Vegas, NV. It is expected  that the
relative importance of cooking emissions in other cities will vary based on regional restaurant
density and the magnitude of other anthropogenic emissions including VCPs and mobile sources.
More work is needed to quantify cooking in other urban areas. Measurements from this study in
Pasadena, CA (Fig S12) and those conducted previously in Livermore, CA and Atlanta, GA show
that long-chain aldehydes are ubiquitous in urban air (~100 – 200 ppt) and modulated by
commercial and residential cooking (Wernis et al., 2022;Peng et al., 2022). The source
apportionment profiles determined here may be compared against other urban environments to
evaluate cooking in other cities.

The VOCs emitted from cooking are reactive and may contribute to the formation of ozone,
secondary organic aerosol, and other pollutants such as peroxyacyl nitrates (Bowman et al., 2003).
A review of VOC emissions inventories show that total cooking emissions (i.e., residential +
commercial cooking) are likely underrepresented in air quality models. Spatial patterns of long-
chain aldehydes suggest that more work is needed to quantify the magnitude of emissions from
commercial cooking, which are also important sources of primary urban SOA (Robinson et al.,
2018;Shah et al., 2018). PMF results in Las Vegas suggest that cooking emissions may be as
important to urban VOCs as mobile sources in regions with significant restaurant activity.

**Data Availability**

Data for SUNVEx and RE-CAP are available at the NOAA CSL data repository
(https://csl.noaa.gov/projects/sunvex/).

**Author Contribution**

MMC, CES, XL, JBG, AL, JP, HJB, KA, and CW conducted measurements during SUNVEx and
RE-CAP. CH, QZ, RHS, JH, ML, KS, and BC developed inventories used to compare against
observations. MMC and CW wrote the paper with contributions from all authors.

**Competing Interests**



The authors also have no other competing interests to declare.
**Acknowledgements**

MMC, CES, QZ, and RHS acknowledge support from the U.S. Environmental Protection Agency
(EPA) STAR program (grant # 84001001). The views expressed in this document are solely those
of the authors and do not necessarily reflect those of the Agency. EPA does not endorse any
products or commercial services mentioned in this publication. CW, MMC, CES, LX, JBG, and
AL acknowledge measurement funding from Clark County, NV (contract number 20-022001) and
the California Air Resources Board (contract number 20RD002). This work was supported in part
by the NOAA Cooperative Agreements with CIRES, NA17OAR4320101 and
NA22OAR4320151. The authors thank Paul Wennberg, John Seinfeld, and Ben Schulze for their
coordination of the Caltech ground site during RECAP-CA.

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
