# Peer review of "Contribution of Cooking Emissions to the Urban Volatile Organic Compounds in Las Vegas, NV"

_EGUsphere, 2023_

## Author Comment (AC1)

We thank the reviewers for their helpful comments and efforts to improve the manuscript. Our responses to reviewer comments is provided below.

We note the following key changes to the manuscript:

(1) Based on reviewer comments, we have included a supplemental figure showing a map of each drive to highlight the locations and time-of-day of the mobile laboratory sampling.

(2) Based on reviewer comments, we have included a comparison of CO with the mobile source factor in Fig. 8.

(3) Based on reviewer comments, we have included a supplemental table with PMF factors for the 5-factor solution.

(4) We have separated the campaigns descriptions of SUNVEx (Las Vegas study) from those of RECAP-CA (Los Angeles study) and supplemental mobile drives.

(5) We have included table of contents to the supplement to help organize the supplement.

(6) In revising the manuscript, we have also updated the restaurant density data shown in Figures 1, 3 and 4. In our previous figures, we included data for all restaurants outlined in South Nevada Health District (SNHD) reports accessed in December, 2021. We believe that it would be more accurate to only present data for restaurants that were inspected within one year of the SUNVEx campaign (i.e., after July 1, 2020). This criteria is consistent with the minimum frequency that SNHD conducts inspections (at least once per year). The trends in restaurant density are identical to our previous plots, but the total number of restaurants decreased by ~30%. The total number of restaurants in Clark County resulting from this screening is ~5020, which is consistent with total restaurants + food trucks gleaned from online tools (e.g., TripAdvisor, 5300). We note that this only affects the results presented in Figures 1, 3, and 4.

We now include a brief description of how the restaurant data from the SNHD are screened in in the main text (lines 134-137), and then provide a more complete description in the SI (Section S2). This provide the reader with a description of how the data were processed and how much restaurants reflect total commercial food activity in the Las Vegas region.

**Reviewer 1:**

The article from Coggon et al., investigates the contribution of cooking emissions to the volatile organic compounds measured in the urban area of Las Vegas, NV. The interest in the subject stands from the need to improve air quality in urban areas through a better characterization of

present pollution sources. To my knowledge this is the first field study focusing on volatile organic compounds emitted from cooking. The authors identified some marker compounds using robust experimental methods (PTR-TOF-MS coupled with GC), compared cooking emissions with existing inventory emissions, and apportioned the cooking source using a positive matrix factorization analysis. I find the research outcome of great value and of interest to the atmospheric chemistry research community and in scope with the journal. I definitely recommend the article to be published, and only have some minor comments to make some points more clear to the readers.

1. L132. How was the sampling with the mobile laboratory conducted? How were the local emissions from the mobile lab excluded from the total VOC measured in air? A few lines to describe the sampling and how the focus drives were organized would be of interest.

   Thank you, we now provide a separate paragraph to explain the mobile laboratory sampling strategy. We also include the mobile laboratory drive path(s) in Fig. 1 and to Fig. S1 to highlight the different regions sampled during mobile lab. The inlets of the mobile laboratory are located on the roof towards the front of the van and away from the mobile laboratory exhaust. We primarily analyze data when the mobile laboratory is in motion and least likely to self-sample its own exhaust. When operating as a stationary site, the engine is turned off and instrument power is maintained by a series of batteries charged using an inverter/charger. The following text has been added:

   "The mobile laboratory sampled air in regions across Las Vegas to investigate anthropogenic emissions from residential, commercial, and entertainment districts (Fig 1). A full description of the mobile laboratory is provided by Eilerman et al. (2016) Briefly, instruments sampled air from inlets located on the roof towards the front of the vehicle. Data are analyzed when the mobile laboratory is in motion in order to eliminate periods when instruments might self-sample exhaust. When operating as a ground site or during stationary sampling, the mobile laboratory engine is turned off and instruments are powered by batteries that are charged with a MagnaSine inverter/charger (MS2812). Pressure, temperature, relative humidity, and wind speed / direction were monitored by a suite of meteorological sensors (Airmar 200 WX, R.M. Young 85004 sonic anemometer). Mobile laboratory position, speed, and heading were measured by a differential GPS (ComNav G2B)."

2. L213. How were fragments considered in the final quantification of the molecules reported here? Is the dominant fragment considered or a sum of all the identified fragments?

   We only consider the proton-transfer product for final quantification. The dehydration products (m/z 111 for octanal and m/z 125 for nonanal) have potential interferences from cycloalkane fragments that could bias these measurements in regions with significant mobile source or natural gas production (e.g., Coggon et al. 2024, Gueneron et al. 2015). We now note this in the text with the following:

"We note that mixing ratios of nonanal and octanal are quantified based on the signal at the proton-transfer product and not from the sum of all fragments. The fragmentation products from octanal, nonanal, and other aldehydes overlap with signals from cycloalkanes emitted from fossil fuels and biogenic isoprene (Coggon et al., 2024; Gueneron et al., 2015)."

3. L230. The authors use PMF analysis on two specific periods of the field campaign. Is there a particular choice of these 2 periods (meteorology or other changing factor)?

These two periods were chosen due to overlap with other instrumentation. Furthermore, there was a ~10 day issue when the air conditioner in the ground site failed and temperatures in the enclosure reached >65C. Amazingly, the instrument survived, but the data quality was quite poor.

We now clarify in Section 2.1 that data were not available between July 9 – 19 due to instrument issues (line 130). At line 248, we note that the two periods correspond to when PTR-ToF-MS measurements were available.

4. L482-487. How was the adjustment done ? Could you be more specific or rephrase this comment ?

This adjustment was performed by dividing the VCP and mobile source factors by the fraction of mass determined by Gkatzelis et al. to be measured by the PTR-ToF-MS – i.e., VCP_adjusted = VCP_PMF / 0.5 and Mobile_adjusted = Mobile_PMF / 0.47. We now update the text to be more specific:

"The mobile source and VCP-dominated factors shown in Figures 9 and 10 have been adjusted by the following equation to account for this unresolved mass.

$$M_{total} = \frac{M}{a}$$

$M_{total}$ is the adjusted PMF factor, $M$ is factor mass, and $a$ is the fraction of the factor mass estimated by Gkatzelis et al. (2021) to be measured by PTR-ToF-MS for VCP (0.5) and mobile source (0.47) emissions."

5. L502. Could you provide more information on how you normalized the diurnal profiles to the impact of meteorology?

We apologize, this phrasing does not quite capture what we are trying to convey in Figure 10A. We aim to show how the distribution of the PMF factors vary as a function of time of day, and by normalizing to the PMF sum of anthropogenic factors (mobile source, VCP, cooking), we believe this helps to visual the results without the large meteorological effects observed in Fig 9. We have now revised the text with the following:

"Figure 10A shows the fraction ( $f_i$ ) that each primary factor contributes to total VOC mass resolved by PMF at the Jerome Mack ground site.

$$f_i = \frac{M_i}{\sum M_i}$$

Where $M_i$ is the mass concentration of the VCP-dominated, mobile source, and cooking-dominated factors. Here, the denominator represents the total anthropogenic emissions resolved by PMF. The local solvent factor is excluded from this analysis since it is not representative of regional VOC concentrations. Figure 10B shows the average factor contribution over the entire analysis period."

6. L512-513. What are the main sources and activities emitting these compounds? Are they also associated with personal care products consumption?

Yes, these emissions are largely driven by personal care products along the Las Vegas strip (as indicated by FIVE-VCP). We now note this in the text.

7. L597 How is the population density, number of restaurants and other activities differing between the two areas?

This is an excellent question, and likely dependent on variations in population density associated with tourism, potential differences in VCP use, etc. For example, in July 2021, approximately 3.3 million people visited Las Vegas (or ~106,000 people/day). Along the Las Vegas strip and downtown area (where most tourism occurs), the restaurant data indicates that there are ~550 restaurants. If we assume a population of 106,000 people along the strip, then this would translate to ~530 restaurants / 100,000 people. If we sum all of the restaurants in Clark County (5,020) and divide this by the total population + tourism (2.4 million), then we observe an average restaurant density of ~210 restaurants / 100,000 people. So this may suggest that cooking / VCP usage along the Strip is twice as high as Clark County as a whole.

We have included the following discussion at lines 655 – 664:

"Restaurant statistics indicate that the Strip and downtown regions of Las Vegas have ~550 restaurants (Fig. 1). In July 2021, approximately 106,000 tourists visited Las Vegas every day (LVCVA, 2024). Assuming that the tourism population dominates in this region, this would suggest that there are ~530 restaurants per 100,000 people within the entertainment districts.   In contrast, there are ~5000 restaurants and 2.4 million (including tourists) in Clark County. This indicates that there are ~210 restaurants per 100,000 people county-wide. This suggests that the ratio of cooking / VCP emissions along the Las Vegas Strip may be more than twice as high as those in Clark County. These difference do not account for other factors that may affect emissions, such as the styles of cooking conducted in each region."

8. SI: "In all figures and tables, we correct aldehyde sensitivities using the carbon-dependent fragmentation patterns shown in Fig. S2. First, we determine PTR-ToF-MS sensitivities using

measured or estimated proton transfer rate constants as described by Sekimoto et al. (2017). We then multiply this sensitivity by the fraction of total signal attributed to the proton-transfer rate constant of the aliphatic aldehyde with the same carbon number."

Could you add formulas to help following this part?

We have added formulas as suggested. The new text reads:

"In all figures and tables, we correct aldehyde sensitivities using the carbon-dependent fragmentation patterns shown in Fig. S4. First, we determine PTR-ToF-MS sensitivities ($Sens_{est}$) using measured or estimated proton transfer rate constants as described by Sekimoto et al. (2017). We then multiply this sensitivity by the fraction of total signal attributed to the proton-transfer rate constant of the aliphatic aldehyde with the same carbon number ($\alpha$)."

$$Sens_{Corr} = Sens_{est} \cdot \alpha$$

$$\alpha = \frac{S_{H3O}}{S_{H3O} + S_{Water\ Loss} + S_{Other}}$$

9. Fig. S8: Could you add a table with the mass list for each factor from the 5 factors solution adopted? Similarly to what is done for the factor cooking in Table S1.

We have adopted this suggestion and now include this full table of each factor profile in the SI.

10. Table S1. What is the individual number mass fraction and total mass in the table referring to? How are the unspeciated assessed from the total mass?

The individual mass fraction reflects the ratio of the mass to the total PMF profile derived by PMF. The "totals" reflect the sum of each sub-table above – i.e., 70% are identified as aldehydes and acids with likely isomers identified, 5% are "other masses (lower certainty)" with the likely functionality noted (acid, aldehyde), and 25% are "unidentified" masses – i.e., the balance of the total fraction identified (= 1-[0.7+0.05]) These include hydrocarbon masses that may reflect alkenes + fragments as well as other mases that are typically treated as VCPs, but were partially attributed the cooking factor (e.g., siloxanes). These "mixed" masses reflect a small (<2%) fraction of the total mass.

We now include these details in the Table caption.

**Reviewer 2:**

This study aims to evaluate the spatial distribution and contribution of cooking emissions to VOCs. The authors measured VOCs using PTR-ToF-MS through mobile laboratory and ground site measurements in Las Vegas, Nevada, which was the main study location. Supplemental observations were also conducted in Los Angeles, California and Boulder, Colorado. The mobile laboratory observed a significant enhancement of nonanal and octanal while parked downwind of restaurants in Los Angeles and Boulder. The species were also observed in Las Vegas during mobile measurement, and their spatial distribution was established. Moreover, this study estimates the contribution of cooking emissions' impact on urban air quality through source apportionment using PMF analysis.

The topic is relevant and important to advancing our knowledge in atmospheric chemistry, particularly the role of cooking in air quality. The combined mobile and ground measurements approach is interesting and resulted in valuable insights into the spatial distribution of cooking emissions. I have a few concerns about the approaches, results, and discussion as laid out in the comments below. Additionally, there are a few technical errors and suggestions to improve the legibility of the paper.

Overall, I recommend the acceptance of this manuscript for publication in the Journal of Atmospheric Chemistry and Physics after minor corrections.

1. It's explained that seven mobile measurements were conducted (Lines 128-130). Can this be elaborated, such as how long each drive was, what day and time, did each drive cover the whole Las Vegas area, etc.? Adding this to the Methods section will provide context when discussing observation results.

   Thank you for this suggestion. We have included more details about the mobile laboratory and sampling strategy in Section 2.1. We have also included a supplemental figure (Fig. S1) that highlights the dates, times, and drive paths for each drive. The following text has been added to describe the mobile laboratory measurements:

   "Seven mobile laboratory drives were conducted between 27 June and 31 July, 2021. Supplemental Figure S1 shows the individual drive paths on each day. Drives times ranged between 4 – 10 hr long and the cumulation of the sampling paths provided a nearly-complete survey of the Las Vegas Valley and surrounding desert ecosystem (Fig. 1). Most drives included focused sampling along Las Vegas Strip due to its significant influence on the spatial distribution of VOCs in the Las Vegas Valley. The Las Vegas Strip was sampled at different times of day (afternoon: 12:00 – 18:00 PM, evening: 18:00 – 22:00 PM, night: 22:00 PM – 02:00 AM) to investigate changes to anthropogenic VOC mixing ratios as a result of increased dining, gambling, and entertainment activities in the evening."

2. I am not sure what the purpose of the LGR CO instrument (Line 221) is. I don't see a discussion about CO measurements in the subsequent sections. Was CO not measured at Jerome Mack and by mobile lab? CO is a primary emission tracer. Comparing CO and VOC measurements helps identify primary VOC and secondary factors.

Thank you for addressing this. In earlier versions of our manuscript, we included VOC/CO ratios and estimates of octanal and nonanal emissions. This discussion was removed since it was largely dependent on data from LA.

We agree with the reviewer that CO is helpful in identifying primary VOC factors. We have adopted the reviewer comment related to the PMF-factor comparison and provide a discussion of these comparisons (see response to comment 5)

3. I would expect cooking emissions, including nonanal and octanal, to be elevated when the mobile lab was sampling stationary near a restaurant (Fig. 2; Lines 255-258). What I am curious about is if there was any event when the nonanal and octanal were enhanced while the mobile lab was driven past restaurants? If the instrument could not detect nonanal and octanal because the vehicle emissions are dominating, the VOC spatial distribution in Fig. 3, would need to be adjusted for the vehicle emission background.

Yes, there are periods when octanal and nonanal were elevated while passing by restaurant; this is very similar to the behavior of our stationary measurements (the mobile laboratory is downwind of these restaurants at street level). What Figures 3 and 4 are intended to show is that restaurant density drives the spatial variability observed across the Las Vegas Valley. This behavior is consistent with mobile laboratory measurements of organic aerosol conducted in other US cities (e.g., Oakland, CA, Shah et al. 2018; Pittsburgh, PA, Robinson et al. 2018).

We do not believe a motor vehicle emissions are significantly contributing to the variability observed here. Figure 2 is intended to highlight this point. First, this figure shows that while driving around regions with high population density and significant traffic (e.g., Los Angeles), our only observations of significant aldehyde enhancements occurred downwind of restaurants. Second, we are able to rule out significant impact from tailpipes on the observed spatial variability because we do not observe strong enhancements (or correlation) of octanal and nonanal with traffic markers, such as benzene (lines 282 – 286 and Fig 2). This does not suggest that traffic is a non-zero source of octanal / nonanal, but it does support previous studies which show that on-road long-chain aldehyde emissions from US motor vehicles are small (Gentner et al. 2013).

4. The late-night drive at 21:30-01:00 looks similar to the midday 12:00-19:00 drive (Fig. 4; Lines 317-322). The late-night pattern may look more significant because the scale in Fig 4C differs from those in 4A and 4B. I agree that the mixing ratios can be lower during the midday drive due to a higher boundary layer during the day. If the focus is on the transient events rather than the absolute value, all panels can be in different scales to emphasize them.

We recognize that this wording is confusing. Our focus here is on the broad-scale enhancements observed along the Las Vegas Strip. Aldehyde mixing ratios during the evening drives are more sustained / homogeneous across the region compared to midday drives where the mixing ratios are more variable (albeit, of similar magnitude at times, likely due to observations of concentrated plumes). We interpret this to be a possible combination of emissions (fewer restaurant emissions during midday drives, more during evening / night) and meteorology (less build-up of emissions during the day, greater buildup into a nocturnal boundary layer at night).

We have rephrased this section to the following:

"Nonanal mixing ratios are high and sustained along the Las Vegas Strip during the evening drive when activities from the entertainment industry, including dining, are likely most frequent (panel B). Enhancements in nonanal are also observed during the mid-day and late-night drives (panel A + C), though mixing ratios appear more variable"

5. Restaurant plume observations were done in LA and Boulder instead of Las Vegas. Is there particular reasons why it wasn't done in Las Vegas? The text implies that downtown Las Vegas is dominated by hot dog restaurants emissions because of the high R^2 (= 0.82) with LA hot dog shops. This high correlation seems coincidental. Comparison with laboratory meat cooking (Klein et al., 2016) is interesting as it suggests a broader/generic source group (Lines 389-392). However, there is no report of the R^2 of comparison between this study and Klein et al. Additionally, I would expect time series comparison with other measurements, such as CO (related to Comment 1b), for PMF factors characterization. Are there no standard air quality parameters monitoring at Jerome Mack station?

We identified the role of restaurant emissions after completing the Las Vegas study. The LA and Boulder measurements were conducted to assist in interpreting our results. We do not anticipate that fast-food emissions measured in LA or Boulder would differ from those in Las Vegas.

We agree that the high correlation is coincidental. It is also quite surprising that a region with a diversity of sources such as the Las Vegas Strip would have a mass spectrum that so closely resembles that of a hot dog restaurant. Our intention with this comparison is to show that the mass spectrum in large region of Las Vegas is indeed consistent with a mass spectrum of a restaurant, and possibly associated with meat cooking. We recognize that this may be a distraction and not necessary for the comparison. We have removed the comment about the $R^2$.

We do not make direct comparisons against Klein et al. as these mass spectra data are not readily available in the published materials. Klein et al. provide bulk statistics about the aldehyde / acid content; consequently, we compare the bulk aldehyde / acid content of our Las Vegas measurements with those of Klein et al., Schauer et al., and our individual restaurants

(Fig. 7). Overall, these comparisons support that our observations are consistent with cooking emissions.

CO measurements were available at the site, though these were not an initial focus of our study. We agree with the reviewer that these are useful and now include a comparison of CO to the PMF results in Fig. 8. Broadly, CO is best-correlated with the mobile source factor ($R^2$ = 0.72), followed by the VCP-dominated factor ($R^2$ = 0.67). As the Reviewer notes, these correlations support that these sources are mostly driven by primary sources. We note that the correlation of VCPs with CO likely reflects that VCPs and mobile sources are emitted over coincident temporal and spatial scales. This has been a conclusion in our previous work on VCPs and mobile source emissions (Coggon et al. 2018, Gkatzelis et al. 2021a, Gkatzelis et al. 2021b) and we now provide this discussion in the main text (lines 473 – 480).

6. VCP-dominated factors were adjusted for unresolved mass, while other factors were not (Lines 486-489). What kind of adjustment and how was the adjustment (or no adjustment) implemented for these factors?

   Thank you, we agree this requires clarification. Please see our response to Reviewer 1, comment 3. We now provide additional details and an equation to show how this adjustment as implemented.

7. Line 110: "made in"?

   Thank you, this has been corrected.

8. Lines 137-138: The purple shades are scattered around Las Vegas and not only cover the Las Vegas Strip/Las Vegas Blv. The legend of Fig. 1 describes purple shades as an entertainment district. I'd suggest rephrasing/clarifying this part.

   Thank you, we now refer to the Las Vegas Strip as "the purple shaded region in the center of Fig. 1"

9. Fig. 6: Error in the title of panel B.

   Thank you, this has been corrected.

10. Line 439: Fig. S4 doesn't show the composition of the mobile source factor. It shows the aromatic species measured at Jerome Mack.

    Thank you, this now points to the correct figure, S8.

11. Line 445: Explain PCBTF and other acronyms that haven't been explained.

    This has been corrected to spell out parachlorobenzotriflouride

12. Lines 447-449: I'd recommend adding a label for PCBTF in Fig. 8 Local Solvent factor profile. At the current state, it isn't clear in Fig. 8 how PCBTF is an important and main driver of the Local Solvent factor.

13. Lines 482-486: This is a very long sentence and will lose the audience mid-sentence.

    This has been shortened to read "Gkatzelis et al. (2021) found that ~53% of mobile source emissions and ~50% of VCP emissions are associated with molecules that cannot be resolved by PTR-ToF-MS (e.g., alkanes and small alkenes)."

14. Line 503: "contributes to total VOC"?

    Thank you, this has been corrected.